# Participatory Development of Digital Innovations for Health Promotion Among Older Adults: Qualitative Insights on Individual, Contextual, and Technical Factors

**DOI:** 10.3390/ijerph22081311

**Published:** 2025-08-21

**Authors:** Katja A. Rießenberger, Karina Povse, Florian Fischer

**Affiliations:** Bavarian Research Center for Digital Health and Social Care, Kempten University of Applied Sciences, 87437 Kempten, Germany

**Keywords:** games for health, gamification, serious game, co-creation, participation, gerontology, digitalization, skill, context, ethnography

## Abstract

Location-based games offer innovative approaches for health promotion among older adults, but their effectiveness depends on understanding complex contextual factors beyond technological design. In our study, we aimed to adapt a location-based game in the form of a smartphone application which originally targeted younger people. We employed ethnographic observations in a field test under real-world conditions for identifying the needs and preferences of older adults in this regard. Field notes of one co-creative workshop were analyzed using thematic analysis. Four key contextual factor categories emerged that significantly influenced user engagement: (1) temporal/spatial factors including weather conditions, topography, and traffic safety that impacted screen visibility and cognitive function; (2) virtual-physical orientation challenges requiring high cognitive load to transfer abstract digital maps to real environments; (3) individual factors such as technical competence, mobility levels, and prior accessibility experiences that shaped usage patterns; and (4) social dynamics that provided motivation and peer support while potentially creating exclusionary practices. Successful digital health innovations for older adults require a socio-technical systems approach that addresses environmental conditions, reduces cognitive transfer demands between virtual and physical navigation, leverages social elements while preventing exclusion, and accounts for heterogeneity among older adults as contextually interactive factors rather than merely individual differences.

## 1. Introduction

Medical, technical, and social developments have continuously contributed to the higher life expectancy we experience in Western countries today [1]—although health inequalities still remain [2] and stagnating trends have been observed in the past years [3]. However, from 1949 to 2021, the average life expectancy at birth in Germany increased by 14.9 years for women and 13.9 years for men. In 2020, the average life expectancy for women was 83.4 years and for men 78.5 years. According to these projections, the average life expectancy could thus rise to 88.2 years for women and 84.6 years for men in Germany, by 2070 [4]. On the one hand, the years gained give us additional potential to shape our own lives in old age. On the other hand, these additional years of life are not necessarily lived in full health, which emphasizes the need for coordinated activities of health promotion and support when navigating through health-related challenges in old age, which span multiple sectors and stakeholders [5].

The growing number and proportion of people in old age require adequate information and support within the complexity of health and care issues. In this context, health literacy has become an important topic: it refers to an individual’s ability to participate in the healthcare system and maintain good health. Therefore, several competencies are needed to find/access, understand, appraise, and apply health information [6]. Previous studies have shown that older people are particularly vulnerable in this regard, due to their low levels of health literacy [7] and hence resulting in adverse health [8]. A study among the general population in Germany highlighted challenges in the domains of disease prevention and health promotion (e.g., finding information on health promotion in one’s living environment or mental health problems), which can have negative consequences in connection with unhealthy behaviors. In addition, according to the same study, digital health literacy is still only poorly developed. This, inter alia, refers to the assessment of the trustworthiness and neutrality of digital information [9].

Although digitalization has become a central theme in various areas of everyday life, as has the topic of health promotion, particularly since the COVID-19 pandemic [10], the interconnectedness between these areas in old age—in terms of requirements and implications—needs further attention. Digitalization goes along with various promises in innovative formats for health promotion by, e.g., expanding access to health information and health-promoting services, offering the possibility of tailoring interventions, and providing “nudging” approaches towards healthier lifestyles [11].

One of these, in many ways innovative, approaches is location-based games, which are types of games in which contextualized gaming experiences are enabled and allow for gameplay to evolve and progress through the player’s real-world location [12]. These games utilize the location recognition of a mobile device to enable a gaming experience within a specific context of the real environment. This creates a hybrid world: the boundaries between the digital image of the game environment merge with the real world in which the player moves in real time. Although they were initially used for commercial purposes, location-based games have become increasingly interesting for educational use [13]. They can, for example, be used to increase motivation of people to “do” by combining playful elements with educational content and can, therefore, be used to benefit an individual’s health [14].

We understand games as an untapped potential in the context of health promotion [15], as long as they are developed in accordance with the needs of the users while also considering contextual and technical factors. To achieve this, we have opted to apply a participatory design framework to actively involve and include older adults in the development process [16]. With this, participatory design methods, such as co-creation, move beyond traditional user research or product design and testing and are often presented as a means to increase democratization and empowerment within technology development [17]. Despite these promises, there is a lack of common understanding of what participation is and what it is not, and how it can or even whether it should be differentiated from participatory concepts such as co-creation, co-design, co-production, or even user-centered design, resulting in a multiplicity and (non-)differentiation of terms [18,19]. We therefore embraced a plurality of participatory practices, such as co-creation and instead focused our conceptualization of participation on these methods’ overarching core principles of presence, safety, voice, equity, equality, and authenticity [20].

In this exploratory contribution, we focus on the co-creative workshop of a location-based game targeting older people. This workshop serves as a case study and we explore the role of individual, contextual, and technical factors that need to be taken into account to achieve the goal of developing a more accessible version of the current app. We aim to answer the question of how these factors are enacted in location-based gaming with older adults during a participatory design process to further our understanding of such complex and non-linear processes.

## 2. Materials and Methods

### 2.1. Study Context and Design

As part of our research project, we intend to adapt the existing app Nebolus (https://nebolus.net/en/ [accessed on 17 June 2025]) to promote navigational health literacy to the needs of older people while simultaneously supporting local health services (e.g., counselling services) to gain higher visibility. Nebolus is a location-based game following a scavenger hunt style. It guides app users through playful intervention in the form of an overarching story through the respective city. Thereby, users are guided to health-related institutions—which are stations of the scavenger hunt—and are given additional information about these institutions.

Through an adaptation of this existing app, which was originally designed for younger people, we aimed to make it usable and useful for older adults. Applying a participatory design and development approach, four co-creative workshops have been organized with older adults, to develop a more accessible version of the app. In order to be able to meet the needs and wishes of older people as closely as possible with regard to the design of the interface and content, we have chosen a diverse set of participatory methods and accompanied these with ethnographic field observations under real-world conditions.

In one co-creative workshop conducted in summer 2023, we explored how older people (aged 65+) engaged with the existing version of the app. Following Goffman’s [21] theoretical approaches of interactionism and situationism, we offer a detailed in-depth understanding of the way interactions (with each other and the technical system) as well as contexts (or rather its situatedness) shaped the usability of the tested app during one co-creative workshop in particular. Assuming that the way we interact in social settings, such as co-creative workshops, both influences and is influenced by the surrounding and embedded structures and contexts. Hence, we explored structural conditions that surrounded the workshop (e.g., location, weather, organization) and agentic conditions (e.g., communication between participants, individual actions).

Applying ethnographic field observations supported the provision of an in-depth understanding of the social and spatial context of participants in this co-creative workshop and the way the context configured their experiences of digital technology usage. Additionally, the gathered insights informed further development of an easy-to-access version of the app in testing.

### 2.2. Data Collection

Observation of this workshop took place throughout the entirety of the co-creative activity. During the workshop, small groups of two to three people were formed by the participants themselves. The groups were accompanied by a technology support person and a researcher who prepared field notes based on their observations. The ethnographic data collected for this project focused broadly on the interaction between participants, participants and research staff, participants and their surroundings, and participants and the app during testing. An observation guideline was created based on a close reading of the existing literature pertaining to co-creation with older adults and usability of digital technologies and gamification in old age. Through engaging with the literature beforehand, theoretical sensitivity was provided based on Corbin and Strauss [22] and allowed ethnographers to target their observations accordingly. The participation of researchers during observations was limited. For instance, observers occasionally joined in conversations to provide technological support when participants encountered major challenges in using the app. The observations were recorded as ethnographic field notes throughout the entirety of the workshop. Based on these, protocols were created as soon as possible after the observation took place.

The research site of the first co-creative workshop was an educational and leisure center for senior citizens in Kempten, a town in Southern Germany. It is a centrally located public facility especially for older adults and is open to the public during operating hours. As a meeting space for older adults and other interested persons, it offers numerous daily activities, including but not limited to arts and crafts classes, language courses, exercise groups, and technology-related courses and support. These are organized through staff, volunteers, or other external partners of the space. As an external partner, we were allowed to organize a co-creative workshop with interested participants while using the space’s facilities.

### 2.3. Participants

To recruit participants for the co-creation session, we used various information channels, including radio and newspaper announcements, direct mail to local health-related associations, the distribution of flyers in popular spots in the city (cafés, senior meeting points, etc.), and at the educational and leisure center for senior citizens. Participants (*n* = 7) were aged 65+, able to provide informed consent, and had to have a basic interest in technologies. Four identified as women and three as men. All participants were living in or around Kempten, and appeared to belong to diverse socio-economic backgrounds. Two participants attended as representatives of local health-related associations.

### 2.4. Data Analysis

The created field notes were summarized and digitized verbatim by one of the authors and then formed into a protocol. Names of participants were pseudonymized. Thematic analysis was conducted to uncover themes and patterns in interactions and contextual barriers to app use. Braun and Clarke’s [23] six steps to conduct thematic analysis guided our analysis. After (1) reading and re-reading the protocols to familiarize ourselves with the data, we (2) generated preliminary codes through a methodical analysis of the data extracts. Afterwards, the (3) codes were summarized into main themes, which were then (4) revised in relation to the codes, data extracts, and the dataset as a whole. Subsequently, we (5) refined the themes, naming and defining each one, to then (6) produce a report containing specific cases and to link the themes with our research focus. The coding process itself was inductive through review and discussion of the documents.

## 3. Results

When developing more easily accessible technologies, contextual factors must be taken into account. We tested the then-current version of the above-described app in the field to determine which changes need to occur to enhance its accessibility. While initially focusing primarily on technological details, we soon noticed that, especially with a location-based game, contextual factors have an almost immediate impact on usability, accessibility, and the general experience of potential users of such an app. We identified multiple factors that posed challenges for potential users of this location-based game that are directly linked to contextual aspects and ought to be considered when aiming to enhance accessibility. In the following paragraph, we will present these along the lines of the users’ journey during the testing of this app.

### 3.1. Time and Space as Contextual Factors

Looking at the quite evident contextual factors of time and space, it can be noted that these were relevant aspects while experiencing the location-based app. This became especially apparent when focusing on their respective capacities for concentration and receptivity. While observing the participants throughout the digitally led scavenger hunt, it was noticeable that their respective levels of focus, concentration, and receptivity decreased over time. This could be determined based on them having been increasingly unable to memorize details on the respective stations or needing additional instructions on the route or task. This resulted in an increasing need for technological support. We derive parts of this back to the relatively hot and sunny midday weather conditions with an impending downpour. Combined with the sun shining down at a steep angle on the participants’ screen, creating a reflection that made it hard for them to fully see their display, being outdoors highlighted how such contextual factors can influence the participants’ ability to use the app.

While we were aware of potential mobility restrictions among our participants and had planned the route accordingly, it nonetheless resulted in very heterogeneous groups in terms of mobility, which were amplified by Kempten’s undulating structure. This caused an acceleration of mobility differences within each group and, therefore, put additional demands on participants to mitigate their diverse walking and wheeling paces internally—which further diverged their focus away from the app, route, and traffic. With the participants’ focus being occupied by navigating the app, it can additionally be noted that using the app in active traffic (despite being located in pedestrian zones) may lead to potentially dangerous situations. For example, one participant had to be stopped by one of this paper’s authors to keep them from walking into a building’s pillar while looking at the app on their screen. Another incident occurred only moments later, where a different participant had to be made aware that a large delivery van was attempting to park at the very spot the participant had halted to look at their screen. This leads us to believe that the constant change between virtual space for navigation and information, and physical space to reach their proposed destination, is a considerable challenge for participants and will be further elaborated upon below.

### 3.2. Virtual and Spatial Orientation as Contextual Factors

During the search for the stations, the participants were supposed to be supported by the smartphone’s location recognition and the app’s navigation function. Here, we observed dependencies between challenges in orientation on a technical level (navigation within the app) and spatial orientation (physical environment). To use the app as a navigation device leading them to the next station, participants first had to navigate within the app to reach the app’s navigation mode. Navigation within the app required different lengths of orientation time—due to, e.g., pattern recognition—for the participants until they were more familiar with the app. Some complained that they did not feel sufficiently supported by the app’s route guidance. We also observed during interactions within the groups or with the technology support persons that individual participants needed more time to interpret the meaning of what we initially believed to be basic app functions. Examples of this involved inter alia the fact that the “X” in the top right corner closes windows or pop-up messages, or restricted app use during the scavenger hunt as a consequence of consciously or unknowingly denying app access rights to smartphone functions such as GPS tracking.

In the next step, the participants had to transfer the information displayed in navigation mode, on an abstract digital map of the virtual location, to the real environment. Some participants reported increased mental strain after a short time, with Ms. Müller stating “*That required my full concentration*” [excerpt of observation protocol; translated into English by this paper’s authors]. We interpret this to mean that the interaction of these factors, the high transfer performance of information and the simultaneous processing of external stimuli, required a very high cognitive performance. Once the participants had reached the station according to the app’s guidance, they had to reorient themselves in the real environment to identify the station they were looking for. At times, this transfer appeared to be challenging for some participants, especially when, for example, searching for the station’s code to receive more information on the station and the next clue for the station yet to come.

### 3.3. Individual Contextual Factors

Nonetheless, the heterogeneity of participants, not just regarding mobility, had a direct influence on the way the app was used. Individual factors, such as some participants having local knowledge of the city, enabled them to choose different routes more flexibly. This not only made it easier to adjust the groups’ routes according to different mobility levels but also made spatial navigation easier for them because they did not need to check their screens as often as those who did not have the same local knowledge. Additionally, different pre-existing technical skills among participants often led to those who had an easier time using their smartphones to take the lead with navigating through the app on their devices while also influencing them in navigating physically more often. Thus, this resulted in those with lower technical skill levels generally taking on more passive roles—both virtually and physically.

Another factor playing a role in how each individual used the app extended beyond mobility as a physical trait and also included some users having reduced motor skills—making navigation through the app more difficult than necessary (e.g., targeted swipe gestures). Additionally, minor visual impairments prevented some participants from reading the texts without their reading glasses, adding an additional layer of effort when they had to repeatedly switch between putting their reading glasses on and off while navigating through the city with the app.

Furthermore, notions of each person’s individual experiences, expectations, and motivations influenced how they interacted with the app and how they evaluated it. In terms of technical expectations and attitudes, we could, for example, observe how especially those with less technical experience showed an initial response to decline the app’s access requests, such as GPS location data. This resulted in them not being able to fully use the app’s navigation while also lacking the necessary technical skills to return to access settings at a later point to change their initial response, further exacerbating the effects of their lack of technical skills. But even with previous experiences of navigating through an app (such as Google Maps), participants were either confused or even irritated that the app in question did not look and function exactly the way they expected it to, hence requiring them to focus more on unlearning what they already knew to adjust their usage of navigating through the app. How this was evaluated by participants seemed to be related to their individual appropriation processes of new technologies. Those who appeared more receptive to the gamified elements did not voice or display any signs of irritation but rather curiosity by asking questions. Similar reactions could be observed with those who applied a trial-and-error approach while navigating through the app. This stood in stark contrast with those who presented as more insecure in technology use in general while voicing concerns about making mistakes, such as clicking the wrong button or accidentally deleting anything, such as their progress within the scavenger hunt. Their evaluation appeared more critical while also verbalizing and showcasing a higher stress response and larger mental strain.

Besides this, prior experiences with inaccessibility of locations came to light as well. One participant in particular was a wheelchair user. After collecting his wheelchair, the route and location were easily accessible to him. As an adept technology user, he faced hardly any challenges in navigating through and with the app, as well as transferring digital information to the physical environment. Nonetheless, his prior expectations of facing barriers resulted in him identifying barriers that, on closer inspection, were no barriers to him after all. To illustrate, we would like to refer to a vignette from our protocol, which starts at the group’s arrival at their second station of the scavenger hunt:


*“They found the station’s QR code very quickly. […] Mr. Wagner commented that the QR code was not applied accessibly while pointing at it with one hand and putting the phone down in his lap with the other. He indicated toward a step of about 10 cm height and 1.5 m depth that was right in front of the spot where the QR code was attached to the wall. However, Ms. Zimmermann, […] commented that he could still scan the QR code with his smartphone as it was large enough that he didn’t have to stand directly in front of the code. This proved to be true.”*
[excerpt of observation protocol; translated into English by this paper’s authors]

Before Ms. Zimmermann encouraged him to try it, Mr. Wagner had already given up, expecting to face a potential barrier. This specific event exposed how many barriers people with mobility issues face in our society, that even a person as technologically adept and self-sufficient as Mr. Wagner gave up without even trying. For us as researchers, this was a valuable lesson in terms of how previously made experiences and expectations shape a person’s perception of their surroundings and how they interact with it—and, therefore, also with a location-based game while using it.

### 3.4. Social Contextual Factors

Based on the interaction in the respective participant groups, we observed social factors, which we assume influence app usage—especially one using a location-based game approach in a group setting.

Participants inter alia commented that playing the game in a group, rather than on their own, motivated them to stick with the app-based scavenger hunt. Ms. Müller, for example, stated that “*It was fun—the togetherness. I would probably have switched it off on my own.*” [excerpt of observation protocol; translated into English by this paper’s authors]. Working towards a common goal of finding the next station and, in the end, finishing the scavenger hunt together was perceived very positively by the participants. As illustrated in the previous example, the group setting also unlocked an empowering potential among group members, since without Ms. Zimmermann’s encouragement, Mr. Wagner would not have attempted to scan the QR code.

Nonetheless, while the group setting was unanimously described as a positive factor while using the app, we were still able to observe challenging and, at times, even exclusionary practices that occurred through this. While the previously mentioned situation occurred towards the end of the scavenger hunt, Mr. Wagner originally belonged to a different group. As he had to fetch his wheelchair motor, he and his previous group had agreed to meet at the first station of the scavenger hunt. Unfortunately, his group did not wait for him and had already made their way towards the next station. He did not appear to be surprised or affected by this behavior and was welcomed by his new group without hesitation. Only during the debrief of this situation after the workshop was concluded were we able to determine what exactly had occurred. Despite best efforts to mitigate the risk of something like this occurring, and the researchers’ attempts to make Mr. Wagner’s group members wait for him at the agreed-upon location, it was unsuccessful. The messiness of this situation showed that group members and their dynamics are unpredictable. While we highly recommend organizing such interventions in groups, based on the overwhelmingly positive feedback of the participants, facilitators of location-based games need to be aware of the risks of such unpredictability, that, despite best efforts to be as inclusive as possible and mitigate such risks, discriminatory practices in society can still emerge through this process.

## 4. Discussion

This case study highlights the multifaceted nature of interconnected contextual factors that influence older adults’ engagement with location-based games for health promotion which need to be taken into account when developing and applying digital innovations. Based on fundamentals of design thinking processes, participatory technology development is a non-linear, iterative, and solution-based process to tackle complex challenges [24]. While digital health innovations often focus primarily on technological design and user characteristics, our findings demonstrate that a more comprehensive framework is necessary to ensure accessibility and relevance for older populations. The results of this study reveal that the successful implementation of location-based games for health promotion among older adults depends on the complex interplay between temporal/spatial factors, virtual/physical orientation challenges, individual characteristics, and social dynamics. These contextual dimensions extend well beyond the technological features of the application itself.

The physical environment and temporal conditions significantly impacted participants’ experiences with digital health innovations. Particularly for location-based games, weather conditions (e.g., sunlight, temperature, or impending rain) need to be considered [25] since they directly affect participants’ ability to view screens and cognitive function [26]. In addition, the topography of the city and further characteristics of the real-world environment may create additional challenges, particularly for individuals with varying mobility capabilities. This argument contributes to current discussions regarding barrier-free aspects in urban planning [27].

Environmental conditions as well as individual factors such as different levels of concentration, receptiveness, and technical competence—and particularly their interconnectedness—can influence the usage of location-based games. A strong focus on the app and navigation mode while simultaneously moving through the physical environment requires high levels of mental performance to transfer abstract information to the physical environment. This result is consistent with previous research, which emphasizes that older adults show less flexible spatial cues when navigating in a virtual reality environment compared with younger adults [28]. However, another study has highlighted differences in navigation strategies between younger and older persons and their relationship with visuo-spatial ability [29]. Overall, this evidence is important when designing and applying accessible virtual and physical environments for aging populations [30].

Furthermore, the ability and thus the time required for interpretation of and orientation within individual app functions varied. The decline in focus and receptivity over time suggests that location-based games for older adults should incorporate temporal considerations into their design, potentially including shorter routes or more frequent rest opportunities. This temporal dimension is often overlooked in technology development but proved crucial in our study context. In addition, our observations revealed significant cognitive demands placed on participants as they attempted to navigate between virtual and physical spaces simultaneously. The required transfer of abstract digital maps to real-world environments imposed considerable mental strain, as noted by participants. The challenges observed in digital-to-physical transfer suggest that digital health innovations must bridge this interface more effectively, perhaps through augmented reality features or clearer visual cues that connect digital representations to physical landmarks.

Perhaps even more striking was the profound impact of the social context on participants’ engagement with the location-based game. The group setting provided motivation, peer support, and opportunities for collective problem-solving that enhanced the experience. This finding suggests that social interaction may be as important as technological design in promoting continued engagement with digital health innovations. On the one hand, the shared experience in a group has the potential to provide mutual support. On the other hand, it can also pose the risk of increasing existing inequalities when participants do not respond appropriately or adequately towards existing differences. In order to mitigate risks through conscious or unconscious exclusionary behaviors within the groups, we furthermore recommend that a person should accompany these groups and is sensitive towards issues of diversity and inequality. In this regard, specific actions need to be taken to ensure digital health equity [31].

Although this study provides valuable insights into individual, contextual, and technical factors for the use of digitally supported interventions for healthcare practitioners, there are certain limitations to reflect on. What we assume to be the most limiting aspect is that this paper only reports on one co-creation workshop, thus limiting the transferability of the findings. The geographical location of Germany’s south further limits transferability to other countries or regions in which co-creation with older adults is practiced or gamification is applied in the field of health promotion and disease prevention. Further observations within this area and in other regional settings as well as cultural contexts would add useful comparative data. Furthermore, this study only reports on the observations and interpretations of this paper’s authors and does not necessarily represent the voices of co-creation participants in full. Future studies are needed that focus on the perspectives and experiences of co-creation participants to further our understanding and extend our knowledge. Additionally, longitudinal studies with a larger diversity of participants could examine how contextual influences shift over time as users become more familiar with location-based games.

## 5. Conclusions

Our findings demonstrate that successful digital health innovations for older adults must look beyond technology and individual characteristics to address the complex contextual factors that shape user experiences. Location-based games hold promise for promoting health literacy and physical activity among older adults, but their effectiveness depends on careful consideration of temporal/spatial conditions, virtual/physical navigation interfaces, individual factors, and social dynamics. Our findings suggest that the development of digital health innovations for health promotion among older adults requires a participatory and socio-technical systems approach. Specifically, when developing digital innovations, one should (1) consider environmental conditions (weather, topography, traffic) when designing location-based interventions, (2) reduce cognitive load by minimizing the transfer demands between virtual and physical navigation, (3) incorporate social elements that leverage the motivational benefits of group settings while implementing safeguards against potential exclusionary practices, and (4) account for the heterogeneity among older adults not merely as individual differences but as factors that interact with broader contextual elements.

## Data Availability

The data presented in this study are available on request from the corresponding author. It includes field notes of the researcher of a co-creative workshop with seven participants.

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
