# Peer review of "Participatory Development of Digital Innovations for Health Promotion Among Older Adults: Qualitative Insights on Individual, Contextual, and Technical Factors"

_ijerph, 2025, doi:10.3390/ijerph22081311_

Round 1

Reviewer 1 Report

Comments and Suggestions for Authors

Thank you for submitting your article to the IJERPH. The approach presented in your study is very interesting. However, I believe the following points could help to strengthen the work and better highlight your findings:

  1. The article should include a framework that contextualizes the participatory process and the themes identified. (please include an scheme or figures)
  2. These themes from the participatory research should then be related to other concepts from literature and to propose relevant considerations for development (you may contrast them with user-centered design heuristics for apps, for example).
  3. The main findings should be presented with recommendations for future developments, and also showing how they can be adapted to the Nebolus case study (add figures and use cases).

Finally, the paper presents a valuable contribution by applying pervasive games in the health domain for an older adult target group. However, from the perspective of this reviewer, the findings need to be further complemented with the points outlined above. This will result in a more robust work, both in terms of methodology and future applicability.

Author Response

However, I believe the following points could help to strengthen the work and better highlight your findings:

The article should include a framework that contextualizes the participatory process and the themes identified. (please include an scheme or figures)

We have added further elaboration in the introduction section.

These themes from the participatory research should then be related to other concepts from literature and to propose relevant considerations for development (you may contrast them with user-centered design heuristics for apps, for example).

We added information on the multiplicity/non-uniformity of participatory approaches.

The main findings should be presented with recommendations for future developments, and also showing how they can be adapted to the Nebolus case study (add figures and use cases).

From our point of view, it is no benefit for the readers to receiving explicit figures or use cases for the app itself, because this was only used as an artefact for overall issues in this context. Therefore, we provided general recommendations based on our experiences in the discussion section.

Reviewer 2 Report

Comments and Suggestions for Authors

Location-based games offer innovative approaches for health promotion among older adults, but their effectiveness depends on understanding complex contextual factors beyond technological design.

AUTHORS  aimed to adapt a location-based game in form of a smartphone application which originally targeted younger people.

THEY employed ethnographic observations in a field test under real-world conditions for identifying needs and preferences of older adults in this regard. Field notes of one co-creative workshop were analysed using thematic analysis. Four key contextual factor categories emerged that significantly influenced user engagement: 1) temporal-spatial factors including weather conditions, topography, and traffic safety that impacted screen visibility and cognitive function; 2) virtual-physical orientation challenges requiring high cognitive load to transfer abstract digital maps to real environments; 3) individual factors such as technical competence, mobility levels, and prior accessibility experiences that shaped usage patterns; and 4) social dynamics that provided motivation and peer support while potentially creating exclusionary practices.

AUTHORS conclude that successful digital health innovations for older adults require a socio-technical systems approach that addresses environmental conditions, reduces cognitive transfer demands between virtual and physical navigation, leverages social elements while preventing exclusion, and accounts for heterogeneity among older adults as contextually-interactive factors rather than merely individual differences.

The study is interesting and contributes to the literature

I have the following comments for the authors:

1) Usually, in many MDPI journals, the abstract requires mini-titles. Please check this.

2) From lines 76 and 80, the authors try to explain the objectives, but they do so in a convoluted manner and with excessive use of WeChat.

3) In 2.1, please summarize a bit of the features of the App and also add some pictures/printscreens.

4) The results are interesting. However, a more targeted integration of statistics and a more targeted use of editorial tools such as tables and graphs would make them more appealing.

5) I would also like a paragraph on future developments given the innovative nature of the study.

Author Response

1) Usually, in many MDPI journals, the abstract requires mini-titles. Please check this.

We have compared multiple papers from MDPI and IJERPH specifically and have only found a small number of publications which follow this structure. We therefore decided to leave the abstract as is.

2) From lines 76 and 80, the authors try to explain the objectives, but they do so in a convoluted manner and with excessive use of WeChat.

After double-checking this ourselves and asking for advice from native speakers we do not agree with the first part of this statement on our writing style. The second half of this sentence does not make sense unless the reviewer may have confused it with ChatGPT or another AI based software. If this is the case, we want to point out that no AI based software has been used in the creation of this paper.

3) In 2.1, please summarize a bit of the features of the App and also add some pictures/printscreens.

We feel like we have summarized the app features extensively in 2.1. Furthermore, we provided a hyperlink where the app is visible. The printscreen themselves are not in the focus of our manuscript and we therefore did not add any pictures.

4) The results are interesting. However, a more targeted integration of statistics and a more targeted use of editorial tools such as tables and graphs would make them more appealing.

Since this is a qualitative study, we feel the need to bring awareness to reviewer 2 that this study does not and should not entail statistics. We furthermore think it to be counterproductive to add graphs and tables when applying thematic analysis in the manner we have done.

5) I would also like a paragraph on future developments given the innovative nature of the study.

We provided recommendations based on our experiences in the discussion section.

Round 2

Reviewer 1 Report

Comments and Suggestions for Authors

After reading the new version of the article, I see some points that need to be addressed.

  1. These themes from the participatory research should then be related to other concepts from literature and to propose relevant considerations for development (you may contrast them with user-centered design heuristics for apps, for example).

These points are not discussed deeply in the article. Some of the findings are in line with UCD.

  1. The main findings should be presented with recommendations for future developments, and also showing how they can be adapted to the Nebolus case study (add figures and use cases).

Your findings should be related to specific guidelines, as a contribution to the design and development process of apps and games (for that target group). This is important because otherwise, what is the contribution that your work is making? Carrying out a strong methodology for designing and developing software following UCD methodologies for that target group could be pointed out as a contribution; however, how do your findings relate to this?

Author Response

After reading the new version of the article, I see some points that need to be addressed.

  1. These themes from the participatory research should then be related to other concepts from literature and to propose relevant considerations for development (you may contrast them with user-centered design heuristics for apps, for example).

These points are not discussed deeply in the article. Some of the findings are in line with UCD.

> Thank you very much for your comment. We have added a reference to user-centered design in the background section. However, we decided not to go into further details, because we used a co-creative workshop and the paper aims to provide the results of this workshop. The focus of the manuscript is not to compare different concepts of user involvement. 

  1. The main findings should be presented with recommendations for future developments, and also showing how they can be adapted to the Nebolus case study (add figures and use cases).

Your findings should be related to specific guidelines, as a contribution to the design and development process of apps and games (for that target group). This is important because otherwise, what is the contribution that your work is making? Carrying out a strong methodology for designing and developing software following UCD methodologies for that target group could be pointed out as a contribution; however, how do your findings relate to this?

> Thank you very much for pointing to this issue. We appreciate your comment but regret to tell you that we decided not to incorporate it. The reason is that our manuscript is not meant to provide guidelines for co-design processes. Instead, we aim to focus on results of a co-creative workshop. This dies not allow for any such comparisons with guidelines. Instead, we provide results emphasizing the relevance of contextual factors. This is described and discussed within the manuscipt and provides insights for the readers who aim to develop technologies in a similar setting.